# Development of Polymersomes Co-Delivering Doxorubicin and Melittin to Overcome Multidrug Resistance

**DOI:** 10.3390/molecules28031087

**Published:** 2023-01-21

**Authors:** Eunkyung Han, Doyeon Kim, Youngheun Cho, Seonock Lee, Jungho Kim, Hyuncheol Kim

**Affiliations:** 1Department of Chemical and Biomolecular Engineering, Sogang University, 35 Baekbeom-ro, Mapo-gu, Seoul 04107, Republic of Korea; 2Department of Life Science, Sogang University, 35 Baekbeom-ro, Mapo-gu, Seoul 04107, Republic of Korea; 3Department of Biomedical Engineering, Sogang University, 35 Baekbeom-ro, Mapo-gu, Seoul 04107, Republic of Korea

**Keywords:** multidrug resistance, melittin, doxorubicin, PI3K/Akt/NF-kB, polymersome

## Abstract

Multidrug resistance (MDR) is one of the major barriers in chemotherapy. It is often related to the overexpression of efflux receptors such as P-glycoprotein (P-gp). Overexpressed efflux receptors inhibit chemotherapeutic efficacy by pumping out intracellularly delivered anticancer drugs. In P-gp-mediated MDR-related pathways, PI3K/Akt and NF-kB pathways are commonly activated signaling pathways, but these pathways are downregulated by melittin, a main component of bee venom. In this study, a polymersome based on a poly (lactic acid) (PLA)-hyaluronic acid (HA) (20k-10k) di-block copolymer and encapsulating melittin and doxorubicin was developed to overcome anticancer resistance and enhance chemotherapeutic efficacy. Through the simultaneous delivery of doxorubicin and melittin, PI3K/Akt and NF-κB pathways could be effectively inhibited, thereby downregulating P-gp and successfully enhancing chemotherapeutic efficacy. In conclusion, a polymersome carrying an anticancer drug and melittin could overcome MDR by regulating P-gp overexpression pathways.

## 1. Introduction

Multidrug resistance (MDR) is one of the major barriers in chemotherapy [1]. MDR in cancer cells is often related to the overexpression of efflux pump receptors, such as P-glycoprotein (P-gp), which is a family of ATP-binding cassette (ABC) transporter proteins responsible for pumping out exogenous materials from cells [2]. As P-gp actively pumps drugs out of cancer cells, the intracellular concentration of chemotherapeutic agents dramatically reduces [3]. Many studies have been conducted to overcome cancer cells overexpressing P-gp receptors. For example, some studies have used some MDR modulators, such as verapamil [4], gallopamil [5], and tariquidar [6], to inhibit efflux pump receptors and prevent the pumping out of drugs. However, MDR modulators not only elicit toxic effects but also inhibit the intracellular influx of co-delivered anticancer drugs [7,8]. Other studies have suggested another method to overcome MDR through the delivery of large amounts of chemotherapeutic agents by using nanoparticles. Various nanoparticles, such as polymeric NPs, conjugates, micelles, exosomes, and paramagnetic NPs, have been studied as possible delivery carriers for delivering doxorubicin into cells [9]. However, the delivery efficiency of nanoparticles is still lower than expected because endocytosis of nanoparticles not only causes endosomal entrapment or lysosomal degradation but also induces MDR efflux receptors to pump drugs out of cells [9]. Therefore, a new strategy is necessary to overcome the limitation of chemotherapy caused by overexpressed efflux pumps.

Since cancer cells exhibit resistance to chemotherapeutic agents such as doxorubicin and paclitaxel, alternative approaches should be developed using appropriate inhibitors to inhibit P-gp overexpression at a molecular level. P-gp is encoded by *ABCB1*, and the P-gp promoter contains a κB site, which can be activated by nuclear factor kappa B (NF-κB) [10]. Among P-gp-mediated MDR-related pathways, the PI3K/Akt pathway is a signaling pathway commonly activated in cancer. The PI3K/Akt pathway plays an important role in inhibiting cancer cell apoptosis and promoting cancer cell proliferation, invasion, and angiogenesis [4,5]. Recently, studies have focused on the involvement of the PI3K/Akt signaling pathway in MDR. Therefore, many studies have been conducted to inhibit the PI3K/Akt pathway during anticancer therapy [6,7]. Moreover, transcriptional factors, including NF-κB, bind to the promoter region of the MDR gene to initiate the transcription and expression of the P-gp pump. Thus, the NF-κB pathway is involved in the regulation of the MDR1 expression in multiple cancer cells [8,11,12]. These findings suggest that the downregulation of PI3K/Akt and NF-κB pathways can be a novel strategy to improve the efficacy of chemotherapy by overcoming the limitations of chemotherapeutic agents in MDR cells.

Bee venom (BV) is a complicated mixture of various constituents, particularly melittin (40–50% of the dry weight). Melittin is a basic peptide consisting of 26 amino acids [13]. It shows potential not only as a drug but also as a cell-penetrating peptide. Moreover, it considerably inhibits EGF-induced PI3K/Akt phosphorylation in MDA-MB-231 and MCF-7 cells [14]. It also inhibits lipopolysaccharide (LPS)-induced NF-κB activation by preventing p50 translocation via a protein (melittin)–protein (sulfhydryl group of p50) interaction [15]. However, to the best of our knowledge, few studies have been conducted to explore the function of melittin in MDR by downregulating PI3K/Akt and NF-κB pathways. Despite these advantages, its nonspecific mechanism of action can cause adverse effects on normal cells and induce the hemolysis of red blood cells when it is administered through blood vessels [16]. Melittin also elicits cytotoxic effects, resulting in necrotic and apoptotic cell death. To overcome this problem, further studies should explore a new alternative to develop a drug carrier that can safely deliver melittin and anticancer drugs at the same time.

In this study, we developed a polymersome (Dox-Mel PL; Figure 1) composed of poly (lactic acid) (PLA)-hyaluronic acid (HA) (20k-10k) di-block copolymers and loaded with melittin (Mel) and doxorubicin (Dox). Polymersomes are biologically stable drug delivery systems, whose properties, drug encapsulation, and drug release can be easily tuned by applying various biodegradable and/or stimulatory-responsive block copolymers [17]. CD44 receptor is highly expressed in MDR cells such as MCF-7/ADR, Lovo/ADR, K562/ADR, and HL-60/ADR cell lines [18]. Since HA is the most specific ligand for CD44 activation, the targeting of MDR cells through the CD44 receptor is possible [19]. As a result, the outer part of the polymersome (Dox-Mel PL) developed in this study could bind to CD44 in MCF-7/ADR. We experimentally determined the downregulation of NF-κB and PI3K/Akt pathway by Dox-Mel PL to overcome MDR and enhance the anticancer treatment efficacy of doxorubicin in in vitro. Thus, the simultaneous delivery of melittin and a chemotherapeutic agent via a polymersome to MDR cancer cells could overcome anticancer resistance by regulating P-gp overexpression pathways (Figure 1).

## 2. Results

### 2.1. Characteristics of the PLA-HA Polymersome Encapsulating Doxorubicin and Melittin

The developed PLA-HA polymersome encapsulating doxorubicin and mellittin was confirmed through dynamic light scattering and TEM. A PLA-HA polymersome encapsulating doxorubicin and melittin (Dox-Mel PL) was formed through the self-assembly of a PLA-HA copolymer, with an average size of 293.9 ± 17.53 nm (Figure 2A). The TEM image of Dox-Mel PL shows spherical polymersomes, indicating the successful self-assembly of an amphiphilic PLA-HA copolymer (Figure 2B). The loading efficiencies of doxorubicin and melittin in Dox-Mel PL were 54.05% and 84.25%, respectively. The in vitro release profiles of doxorubicin and melittin from Dox-Mel PL showed that 40% and 36% of the encapsulated doxorubicin and melittin were released within the first 24 h, respectively (Figure 2C). Then, approximately 70% of the encapsulated drugs were released for 200 h. The drug release patterns of melittin and doxorubicin were similar since both drugs are hydrophilic.

### 2.2. Intracellular Uptake of Dox-Mel PL

Figure 3 shows the fluorescence images of the intracellular delivery of doxorubicin and melittin into MCF-7/ADR cancer cells using a PLA-HA polymersome. In the case of doxorubicin delivered into MCF-7/ADR cancer cells by using Dox PL, doxorubicin was faintly distributed in the cells (Figure 3, 2nd column) possibly because doxorubicin delivered inside the cells was rapidly pumped out by the P-gp receptor overexpressed in MCF-7/ADR cancer cells. However, when doxorubicin was delivered with melittin (Figure 3, 3rd column), the amount of doxorubicin was relatively higher than that in the Dox PL-treated group. Melittin and doxorubicin were delivered simultaneously in the Dox-Mel PL-treated group. This finding suggested that melittin and doxorubicin were delivered together into the cells, thereby inhibiting the pumping out of doxorubicin by the efflux receptor. Consequently, the amount of doxorubicin in the cells was relatively high. Moreover, MCF-7/ADR cells were pre-treated with HA to block the CD44 receptor to confirm the CD44-mediated endocytosis of Dox-Mel PL. When MCF-7/ADR cells were saturated with HA, the intracellular uptake of Dox-Mel PL decreased compared with that of the HA-untreated group. This result demonstrated that Dox-Mel PL entered the cells through CD44-mediated endocytosis because of the hydrophilic HA part of the polymersome (Appendix A).

### 2.3. Expression of Drug Efflux Receptors and Cell Signaling Pathways

The MCF-7/ADR cell line, which shows anticancer resistance against doxorubicin, overexpresses P-glycoprotein (P-gp), which is an ABCB1 efflux receptor, by approximately 10,000 times more than MCF-7 cells do [20]. P-gp selectively effluxes doxorubicin [21]. To determine whether melittin affects the P-gp expression via the PI3K/Akt pathway, we analyzed the expression of P-gp, pAkt, Akt, and pNF-κB/p65 in MCF-7/ADR cells via RT-PCR and western blot (Figure 4). We found that the P-gp expression in MCF-7/ADR cancer cells was inhibited by melittin in a dose-dependent manner (Figure 4A), suggesting that melittin could inhibit P-gp transcription and/or translation. Since the P-gp expression is closely related to the Akt/NF-κB signaling pathway, we detected the expression of Akt and NF-κB through western blot (Figure 4B,C). The results showed that Dox-Mel PL effectively inhibited Akt phosphorylation (Figure 4B). The extent of the inhibition of the PI3K/Akt pathway was proportional to the amount of the delivered melittin. Similar to Akt and pAkt, pNF-κB was downregulated in MCF-7/ADR cells as the melittin concentration increased (Figure 4C). These results demonstrated that melittin could inhibit the phosphorylation of Akt and NF-κB and ultimately inhibit the transcription and translation of P-gp.

### 2.4. Intracellular Retention of Doxorubicin

Intracellular doxorubicin retention was identified through confocal microscopy and flow cytometry to determine whether the efflux pump activity was decreased in the melittin-treated group (Figure 5). Verapamil was used as a positive control. The intracellular uptake level of doxorubicin immediately after 3 h of incubation was indicated by 0 h (2nd column); in all groups at 0 h, the fluorescence signal of doxorubicin was well distributed in the nucleus (1st column; Figure 5A). Subsequent changes in intracellular doxorubicin concentrations at 1 h intervals showed that doxorubicin remained in the cells longer in the verapamil- (positive control) and Dox-Mel PL-treated groups than in the Dox PL alone-treated group (negative control). After 3 h, doxorubicin was still observed in many cells in the groups treated with verapamil or Dox-Mel PL. At the initial time, doxorubicin was evenly distributed throughout the nucleus and cytoplasm. Over time, doxorubicin was found only in the cytoplasm, indicating that doxorubicin was pumped out of the cell by MDR efflux receptors located in the cell membrane. Doxorubicin decreased over time in MCF-7/ADR cells with strong MDR, and this decrease was faster than that in MCF-7 cells (Appendix A). The percentage of doxorubicin-negative cells (cells with undetectable doxorubicin fluorescence) was measured through flow cytometry to quantitatively analyze the retention time of doxorubicin (Figure 5B). All groups were treated with each sample and incubated for 3 h. Likewise, the verapamil-treated (positive control) group had fewer doxorubicin-negative cells. The Dox-Mel PL-treated group showed a 20–30% reduction in doxorubicin-negative cells compared with that in the Dox-PL-treated group. Therefore, melittin downregulated the efflux pump function and prevented doxorubicin from being pumped out in MCF-7/ADR cells.

### 2.5. Chemotherapeutic Effects of Melittin in Combination with Doxorubicin

To verify chemotherapeutic effects of Dox-Mel PL, The MTT assay was performed in MCF-7 and MCF-7/ADR cells to verify the chemotherapeutic effects of Dox-Mel PL (Figure 6 and Appendix A). The half maximal inhibitory concentrations (IC_50_) of melittin and doxorubicin in MCF-7 cells were 7.16 and 2.38 μg/mL, respectively. Melittin and doxorubicin in MCF-7/ADR cells had IC_50_ of 6.12 and 50.37 μg/mL (Table 1), respectively. These results confirmed that MCF-7/ADR cells had strong resistance against doxorubicin. Furthermore, the CI was calculated to confirm the synergistic effect of doxorubicin and melittin. As shown in Figure 6, some CIs were lower than 1 after the co-treatment of free melittin and doxorubicin, indicating that both drugs synergistically suppressed MCF-7/ADR cell proliferation. In addition, the CIs in the Dox-Mel PL-treated group indicated that both drugs were synergistic in all concentration ranges of melittin and doxorubicin.

## 3. Discussion

In this study, we developed Dox-Mel PL to overcome MDR in cancer cells by blocking P-gp expression with melittin, a major component of bee venom. Doxorubicin is commonly used as an anticancer agent. However, it is resisted by cancer cells and has toxic effects on healthy fast-dividing cells [22]. Melittin, a peptide component of honeybee venom, has a short bloodstream half-life. It causes side effects on normal cells and hemolysis of red blood cells when it is administered via blood vessels. To overcome these problems, we developed Dox-Mel PL composed of poly (lactic acid) (PLA)-hyaluronic acid (HA) (20k–10k) di-block copolymers (Figure 1). The developed Dox-Mel PL drug delivery system could effectively control MDR cancer cells and provide the following advantages: (1) synergistic effect of the co-delivery of doxorubicin and melittin via a biocompatible polymersome; (2) reduced side effects of doxorubicin and melittin; and (3) prevention of drug resistance through P-gp downregulation by melittin. As shown in Figure 1, Dox-Mel PL overcomes MDR by inhibiting P-gp and downregulating the PI3K/Akt/NF-κB pathway. Therefore, doxorubicin can be effectively delivered to MDR cells because of the reduced P-gp activity; it can also reach its action target and kill cancer cells through apoptosis and necrosis [23,24].

Doxorubicin and melittin were co-loaded in a polymersome by using a PLA-HA copolymer. Polymersomes are nanoparticles formed through a self-assembly method by using a synthetic biodegradable amphiphilic block copolymers; they are highly stable because of their tight membrane [25]. Since polymersomes are made via a thermodynamically stable method, various particles, such as micelles, polymeric particles, and polymersomes, can be formed. The morphological characteristics of polymersomes could be roughly predicted in terms of the molecular weight ratio of a hydrophilic polymer ($f=M_{hydrophilic}/M_{total\;polymer})$ [25]. Polymersomes are formed when 0.25<f<0.4. From this point of view, the PLA (20k)–HA (10k) copolymer has a molecular weight favoring the self-assembly of polymersomes in a solution. On the basis of the properties of the PLA-HA copolymer, we inferred how melittin and doxorubicin were loaded in a polymersome. H. Asadzadeh et al. confirmed the dynamics of melittin and its interactions with diverse polymers through molecular dynamics (MD) simulation [26]. Initially, polymers interact with one another, aggregate, and come in contact with melittin. The PLA polymer can form hydrogen bonds with a peptide [26]. Therefore, hydrogen bonds formed between the PLA polymer and positively charged polar residues of melittin; then, they would be embedded in the PLA polymer in a PLA-HA polymersome. Since doxorubicin loading is based on the exchange of ammonium ions [27], doxorubicin would be loaded in the hydrophilic core part of polymersomes. With the advantages of polymersomes, doxorubicin and melittin were safely loaded into polymersomes and sustained the release of the loaded drugs until 200 h (Figure 2).

Anticancer drug resistance is caused by various factors, such as genetic factors (gene mutations), growth factors, increased DNA repair capacity, and enhanced drug efflux [28]. One of the major mechanisms responsible for doxorubicin-induced MDR is the upregulation of the MDR1/P-gp expression. Doxorubicin derivatives are P-gp substrates that can upregulate the MDR1/P-gp expression after repeated exposure of various cancer cells such as breast and lung cancers [29,30,31,32,33]. Moreover, the Akt pathway is actively involved in NF-κB regulation, and NF-κB activity is essential for oncogenic transformation by PI3K and Akt [34,35,36]. Melittin considerably inhibits EGF-induced PI3K/Akt phosphorylation in MDA-MB-231 and MCF-7 cells [14]. Melittin inhibits lipopolysaccharide (LPS)-induced NF-κB activation by preventing p50 translocation via a protein (melittin)–protein (sulfhydryl group of p50) interaction [15]. Figure 3 shows the enhanced doxorubicin delivery efficiency of Dox-Mel PL compared with that of Dox-PL. This result revealed that melittin controlled the P-gp overexpression and showed a synergetic effect with doxorubicin co-treatment. Moreover, Figure 4 illustrates that Mel PL effectively downregulated the P-gp expression and EGF-induced PI3K/Akt/NF-κB NF-κB phosphorylation in MCF-7/ADR cells through RT-PCR and western blotting. After melittin treatment, the phosphorylation levels of Akt and IκBα and the expression of p65 were inhibited, showing that melittin-induced P-gp suppression depends on PI3K/Akt/NF-κB pathway regulation [37]. However, few studies have been conducted to explore the function of melittin in MDR by downregulating the PI3K/Akt/NF-κB pathway. Our results demonstrated that blocking the PI3K/Akt pathway could lead to the downregulation of the MDR1/P-gp protein expression, thereby reversing the MDR. P-gp downregulation was also confirmed through enhanced doxorubicin retention time in MCF-7/ADR cells (Figure 5). When treated with Dox-Mel PL, doxorubicin remained in the nucleus longer than Dox PL, indicating that melittin affected the PI3K/Akt/NF-κB pathway; therefore, the P-gp expression and activity were downregulated.

Although combining two or more therapeutic agents in a single carrier system offers new treatment possibilities, it shows some new challenges, such as changes in release pattern and drug activity. In addition, the clinical development of these co-delivery systems is costly. However, co-delivery system has become a prospective technology for successful cancer treatment [38]. Therefore, we developed a co-delivery system and confirmed whether the treatment effect was increased. To confirm the synergistic effects of doxorubicin and melittin, we analyzed the CI from in vitro cytotoxicity curves. Table 1 presents the IC_50_ of individual drugs. A CI of ≤1 indicates synergistic, additive, and antagonistic effects [39]. The combination of anticancer drugs can work synergistically, additively, or antagonistically depending on the ratio of the combined drugs [40,41]. Figure 6 shows that the synergistic effect was reduced when the amount of the two drugs was excessive. Consequently, the two drugs could create synergies even in small amounts. Therefore, the developed Dox-Mel PL potentially shows an excellent therapeutic effect even in small amounts.

## 4. Materials and Methods

### 4.1. Materials

The following materials were used in this study: doxorubicin hydrochloride (Dox), tetrahydrofuran anhydrous (THF), and dimethyl sulfoxide (DMSO; Sigma Aldrich (St. Louis, MO, USA); melittin (Mel; Genscript, Piscataway, NJ, USA); poly(L-lactide)-NHS (PLA-20K-NHS; Ruixibiotech, China); hyaluronic acid-amine (HA-10K-amine; Creative PEG Works, Durham, NC, USA); RPMI 1640, fetal bovine serum (FBS), and antibiotic–antimycotic (AA) medium (Welgene, Gyeongsangbuk-do, Republic of Korea); and AccuPower PCR PreMix, ABCB1 (P-glycoprotein, P-gp), NF-κB, and PI3K/Akt primers (BIONEER, Daejeon, Republic of Korea).

### 4.2. Preparation of Polymersome

PLA-NHS and HA-amine were, respectively, dissolved in DMSO and distilled water and mixed at a molar ratio of 1:1 to form a di-copolymer. For NHS ester–amine reaction, pH was adjusted to 8 and stirred at 25 °C for at least 24 h. The PLA-HA solution was centrifuged at 4000 rpm for 15 min by using an Amicon Ultra-30 kDa Centricon (Millipore, Billerica, MA, USA) to remove unreacted PLA and HA. Then, the PLA-HA solution was centrifuged at 13,200 rpm for 10 min to harvest PLA-HA copolymer pellets and vacuum-dried in a desiccator for 24 h.

PLA-HA copolymer was dissolved in 2 mL of tetrahydrofuran (THF) and added to the melittin solution dropwise via a syringe pump (NE 4000, KF Technology, Roma, Italy) at 2 mL/h while stirring to allow PLA-HA self-assembly. After 1 h, the mixture was stirred overnight to evaporate THF and centrifuged at 13,200 rpm for 15 min by using a 3k Centricone filter to remove unloaded melittin. Doxorubicin was loaded into the polymersome through an ion gradient method. Doxorubicin hydrochloride (1 mg/mL) was added to 1 mL of ammonium sulfate (pH 5.5, 250 mM). Then, Mel PL was added to the solution and allowed to react while stirring at 25 °C for 2 h. In the case of Dox PL, the PLA-HA copolymer solution was added dropwise to the PBS solution without melittin, and doxorubicin was loaded via the same process.

### 4.3. Characterizations of Dox-Mel PL

The size distributions and zeta potential of Mel PL and Dox-Mel PL were analyzed through dynamic light scattering (Zetasizer Ultra Red Label, Malvern Panalytical, Malvern, UK). The morphological characteristics of Dox-Mel PL were determined using a transmission electron microscope (TEM; JEM-2100F, JEOL, Tokyo, Japan). The in vitro release rate of doxorubicin and melittin from Dox-Mel PL was measured using high-performance liquid chromatography (HPLC; Agilent Technologies, St. Clara, CA, USA). Each group was loaded into a dialysis membrane (Spectra/Por^®^ 7, MWCO 10 kD, SPECTRUMLABS, Los Angeles, CA, USA), placed in 5 mL of PBS solution, and incubated in a shaking incubator at 37 °C. At the predetermined time points, 5 mL of the PBS buffer was replaced with fresh buffer, and the amount of the released drug was confirmed through HPLC For detection of melittin, an analytical HPLC column (Agilent Technologies, USA, 5 µm, C18, 4.6 mm × 250 mm) was used at a flow rate of 1.5 mL/min and gradient time of 30 min, with UV absorbance detection at 220 nm. Mobile phase A was 0.0065% trifluoroacetic acid (TFA) in distilled water, mobile phase B was 0.0065% TFA in acetonitrile. The gradient was linear from 5% B to 80% B for 15 min, after 5 min at 80% B, resetting followed to 5% B in 1 min and re-equilibration for 9 min. The column was thermostated at 45 °C. For detection of doxorubicin, same HPLC column was used at a flow rate of 1 mL/min with UV absorbance detection at 230 nm for 10 min. The mobile phase was acetonitrile and distilled water in the ratio of 50:50 with phosphoric acid 0.6 mL/L and sodium dodecyl sulfate 1.327 g/L. All in vitro release tests were repeated triplet for reliable results.

### 4.4. Confirmation of Gene Expression through RT-PCR

The expression levels of the efflux pump in MCF-7 and MCF-7/ADR cells were analyzed using reverse transcription-polymerase chain reaction (RT-PCR). Cells (1 × 10^6^ cells/well) were seeded in a six-well plate, cultured overnight, treated with Dox-Mel PL and Mel-PL, and incubated for 24 h. After incubation, total RNA was obtained using an RNeasy Mini kit (Qiagen, Venlo, The Netherlands) in accordance with the manufacturer’s protocol. RT-PCR was performed using AccuPower PCR PreMix (BIONEER, Daejon, Republic of Korea). The primers were used as follows: P-gp primer: forward (5’-ATATCAGCAGCCCACATCAT-3’); reverse (5’-GAAGCACTGGGATGTCCGGT-3’), NF-κB primer: forward (5’-CTGGTGATCGTGGAACAGCC-3’); reverse (5’-CAGAGCCTGCTGTCTTGTCC-3’), Akt primer: forward (5’-CAAGAAGGAAGTCATCGTGG-3’); reverse (5’- TCGTGGGTCTGGAAAGAGT-3’), β-actin primer: forward (5’-AGCACAGAGCCTCGCCTT-3’); reverse (5’-CATCATCCATGGTGAGCTGG-3’). β-actin was selected as the internal reference gene; the following reaction conditions were set: pre-denaturation at 94 °C for 5 min; denaturation at 94 °C for 30 s, annealing at 55 °C for 30 s, extension at 72 °C for 30 s, and amplification for 35 cycles; extension at 72 °C for 10 min; and storage at 4 °C. The PCR product was electrophoresed on a 1% agarose gel. Semi-quantitative RT-PCR was determined with a PCR thermal cyclers (T100 thermal cycler, Bio Rad, Hercules, CA, USA) and analyzed using ImageJ (1.53e) (National Institutes of Health, Bethesda, MD, USA).

### 4.5. Confirmation of Gene Expression through Western Blot

The downregulation of the Akt pathway in MCF-7/ADR cells was demonstrated through western blot. Cells (1 × 10^6^ cells/well) were seeded in a six-well plate, cultured overnight, treated with Dox-Mel PL and Mel-PL, and incubated for 24 h. Then, the cells were washed with PBS and lysed with RIPA buffer with a protease inhibitor in an ice bath. The cell lysate solution was placed in an ice bath for 5 min and centrifuged at 13,200 rpm and 4 °C for 15 min. After centrifugation, the supernatant was immediately drawn into a pre-cooled plastic tube, and the concentration of proteins in cell lysates was determined using a BCA protein assay kit (Pierce, Thermo Fisher Scientific, Waltham, MA, USA). Equal amounts of proteins were separated using 10% gels via sodium dodecyl sulfate polyacrylamide gel electrophoresis (SDS-PAGE) and transferred onto polyvinylidene difluoride (PVDF) membranes (ATTO, Tokyo, Japan). After being blocked with EzBlock Chemi (ATTO, Japan), the membranes were incubated with primary antibodies such as Akt, pAkt, pNF-κB (Biorbyt, Cambridge, UK), and GAPDH. The membranes were washed with TBST buffer, incubated with horseradish peroxidase (HRP)-conjugated secondary antibodies, and visualized using enhanced chemiluminescence reagents (Perkin Elmer, USA). The relative levels of each protein were quantified using ImageJ (National Institutes of Health).

### 4.6. Determination of Intracellular Uptake and Retention of Doxorubicin and Melittin

Dox-Mel PL was subjected to fluorescence imaging via confocal laser scanning microscopy (TCS SP8, Leica, Munich, Germany). Cells (1 × 10^6^ cells/well) were seeded in a confocal dish and cultured overnight. The cells were treated with each group in FBS-free media for 6 h, washed with PBS, and stained with a DAPI staining solution. For the intracellular retention of doxorubicin images, each group was incubated in the cells for 3 h; afterward, each group was washed with PBS and further incubated at 1 h intervals. In addition, experiments were conducted using MCF-7 and MCF-7/ADR cell lines under the same conditions to confirm the difference in doxorubicin efflux. Verapamil, known as a first-generation P-gp inhibitor, was used as a positive control.

### 4.7. In Vitro Cell Viability Assay

An MTT assay was performed in MCF-7/ADR cells to confirm the cytotoxic effect of Dox-Mel PL. The cells (1 × 10^4^) were seeded in a 96-well plate and cultured overnight. They were then treated with the following groups: negative control (non-treated), free doxorubicin, free melittin, Dox PL, Mel PL, and Dox-Mel PL. After the treatment, the cells were incubated for 24 h. The MTT solution (0.5 mg/mL) was added to each well, and the cells were incubated at 37 °C for 2 h. Then, 150 µL of DMSO was added to dissolve formazan crystals in living cells. Absorbance intensity was measured at 595 nm by using a microplate reader (Bio-Tek, Winooski, VT, USA). The synergistic effect of the two drugs was verified in terms of the combination index (CI), which was calculated using the Chou–Talalay method in Compu Syn software (1.0) (Combo Syn Inc., Paramus, NJ, USA). Automated computer simulation shows the occurrence of synergism and measures the extent of synergism (CI) at any dose levels (the isobologram) or at any effect levels (the Fa–CI plot) [42].

### 4.8. Statistical Analysis

All experimental data obtained from the cultured cells and animals were expressed as the means ± standard error from at least three independent experiments. The statistical significance of differences between experimental and control groups were determined using Student’s *t*-test. Statistical significance was established at *p* < 0.05, and significant differences were shown by asterisks in the figures.

## 5. Conclusions

In conclusion, our study demonstrated an effective strategy for cancer therapy by using co-encapsulated drugs in a polymersome system to overcome MDR. The developed Dox-Mel PL can deliver doxorubicin and melittin into cancer cells. Therefore, co-delivery can enhance the effectiveness of chemotherapy. Melittin inhibits the efflux of doxorubicin by downregulating efflux pump-related cell signaling pathways, the key feature of MDR cells. As a result, high intracellular doxorubicin concentrations can be maintained, consequently inducing cell apoptosis. Thus, Dox-Mel PL may be a useful strategy to overcome the limitations of chemotherapy in MDR cancer cells.

## Figures and Tables

**Figure 1 molecules-28-01087-f001:**
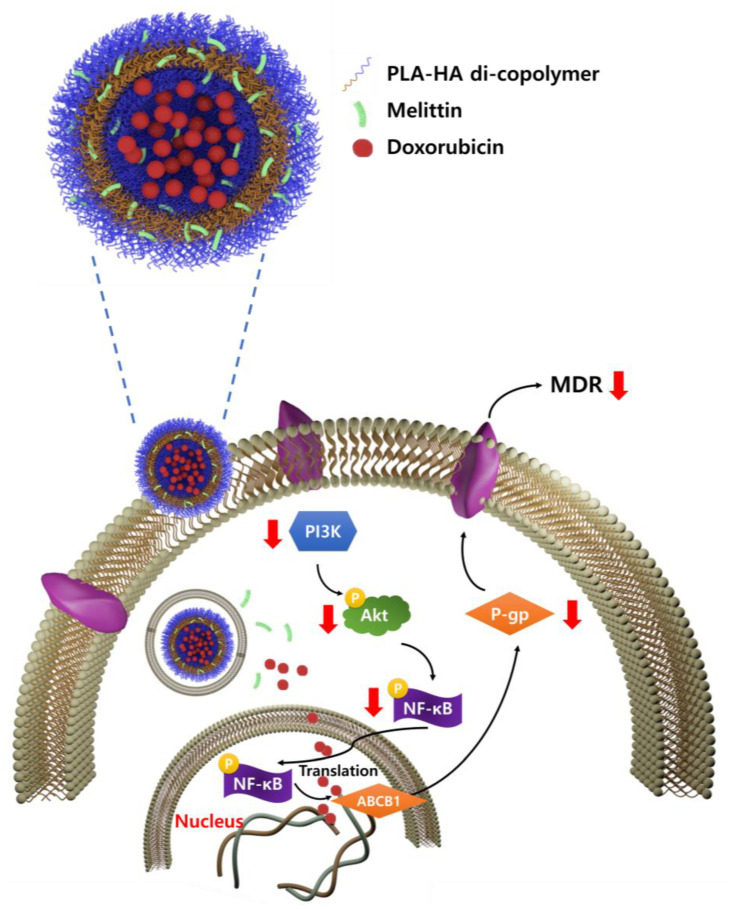
Schematic of Dox-Mel PL demonstrating the co-delivery of melittin and doxorubicin via a PLA-HA polymersome to downregulate PI3K/Akt and NF-κB pathways, consequently overcoming multidrug resistance and enhancing chemotherapeutic efficacy.

**Figure 2 molecules-28-01087-f002:**
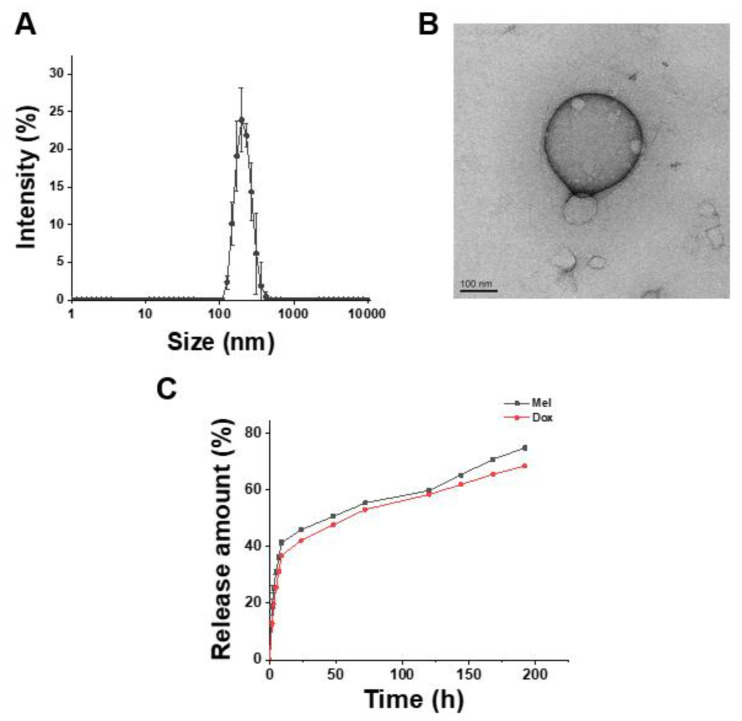
Characteristics of the Dox-Mel polymersome. (**A**) Size distribution of Mel PL and Dox-Mel PL. (**B**) TEM image of Dox-Mel PL. Scale bar represents 100 nm. (**C**) In vitro release profile of doxorubicin and melittin from Dox-Mel PL.

**Figure 3 molecules-28-01087-f003:**
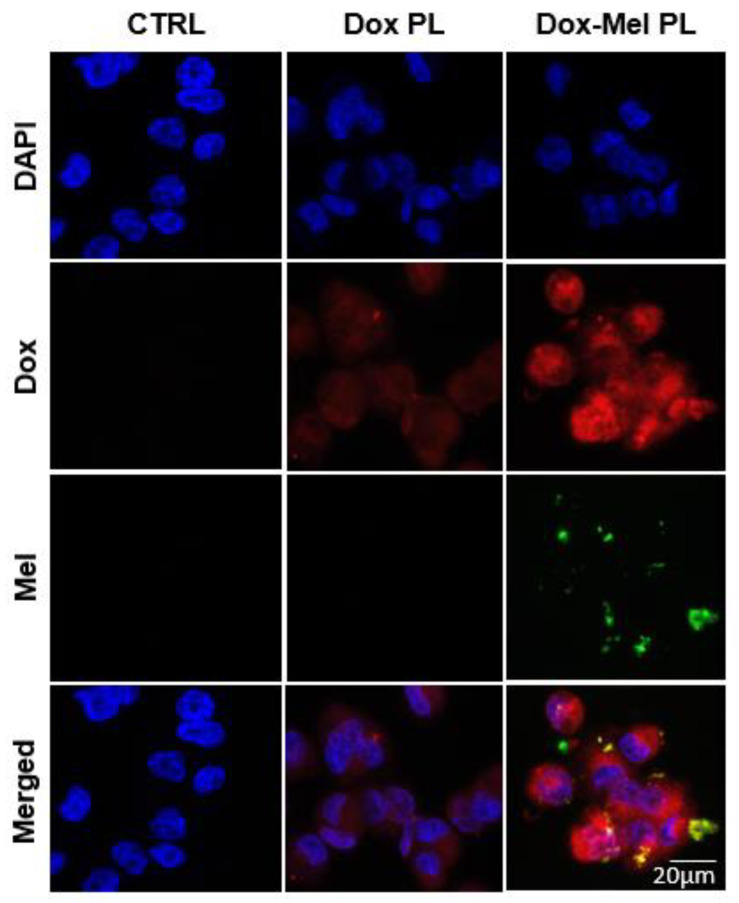
Confocal microscopy images showing the intracellular distribution of doxorubicin and melittin in MCF-7/ADR cells. Cell nuclei are shown in blue. Doxorubicin and melittin are represented by red and green channels, respectively. (Scale bar = 20 μm).

**Figure 4 molecules-28-01087-f004:**
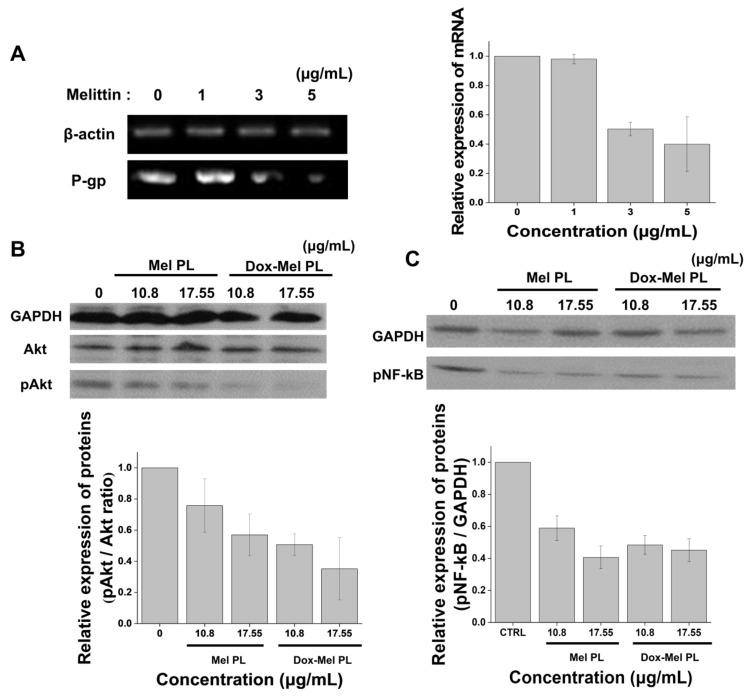
Confirmation of the expression of the efflux pump receptor in MCF-7/ADR cells. (**A**) Semi-quantitative mRNA expression of P-gp in MCF-7/ADR cells through reverse transcriptase RT-PCR. The indicated concentration is based on the melittin concentration. Western blot analysis of the protein expression levels of Akt, (**B**) pAkt, and (**C**) pNF-κB. GAPDH was used as the loading control. The bar charts represent the quantitative comparisons between the groups. The indicated concentration is also based on the melittin concentration. Bar graphs show the mean values calculated from three experiments.

**Figure 5 molecules-28-01087-f005:**
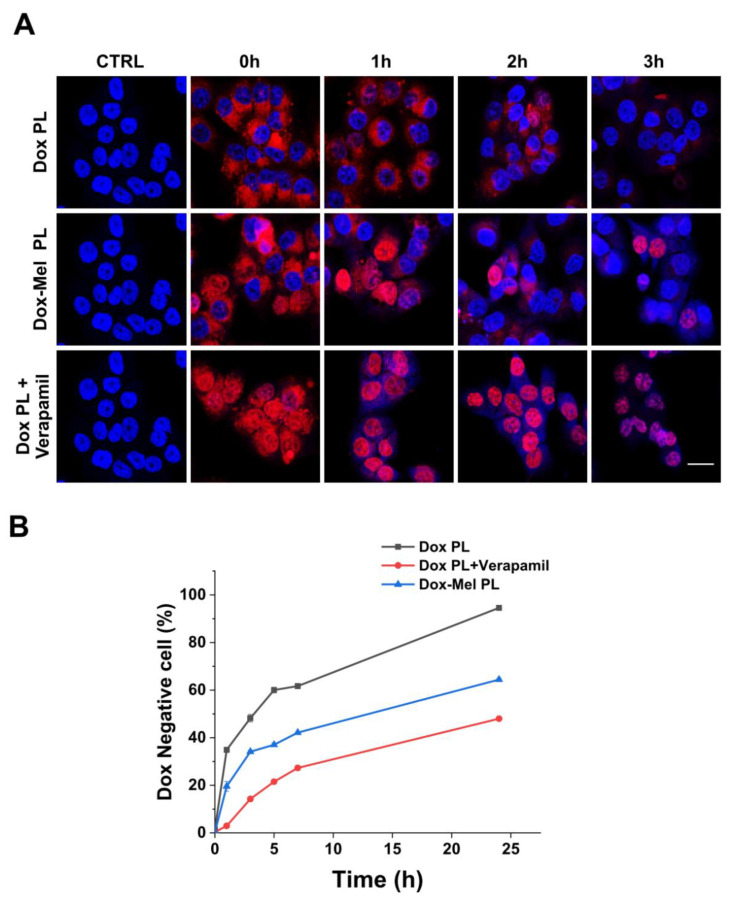
Intracellular retention of doxorubicin. (**A**) Confocal microscopy images showing doxorubicin retention inside MCF-7/ADR cells. Cell nuclei are shown in blue. Doxorubicin is represented by red channels (Scale bar = 20 μm). (**B**) Quantitative flow cytometry data showing the percentage of Dox negative MCF-7/ADR cells.

**Figure 6 molecules-28-01087-f006:**
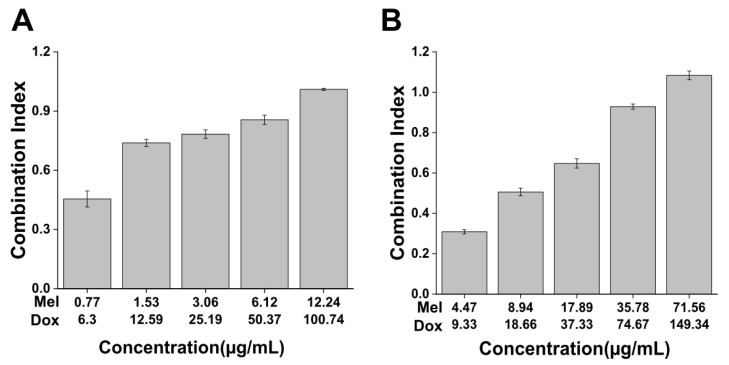
Combination index analysis. Combination index (CI) of <1 indicated that all combinations were highly synergistic against MCF-7/ADR cells. The CI of different ratios of drug combination. Based on IC_50_ of the two drugs, 1/8, 1/4, 1/2, 1, and 2 were treated to experiment. (**A**) Treatment with free doxorubicin–melittin in MCF-7/ADR cells. (**B**) Treatment with Dox-Mel PL in MCF-7/ADR cells.

**Table 1 molecules-28-01087-t001:** IC_50_ of melittin and doxorubicin in MCF-7 or MCF-7/ADR cells.

	IC50 Concentration (μg/mL)	
	MCF-7	MCF-7/ADR
Free melittin	7.16	6.12
Free doxorubicin	2.38	50.37
Mel PL	34.99	35.78
Dox PL	8.65	71.38

## Data Availability

Data are available upon reasonable request.

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
