# Peer review of "Development of Polymersomes Co-Delivering Doxorubicin and Melittin to Overcome Multidrug Resistance"

_molecules, 2023, doi:10.3390/molecules28031087_

Round 1
Reviewer 1 Report
The work submitted for review concerns a very important issue - the ability to overcome multidrug resistance, which is one of the main factors of chemotherapy failure. The authors developed a polymersome made of poly(lactic acid) (PLA-hyaluronic acid (HA) di-block copolymer charged with two substances - melittin and doxorubicin.
The experiment was well planned and executed. In P. 4.3 Characterizations of DOX-Mel PL, it is necessary to provide the parameters of the chromatographic separation - column, mobile phase, flow rate, etc. Whether the release test was performed once or whether the presented results are the average value of several determinations, which should be precisely stated. If it was performed once, the test should be repeated at least twice to make the results reliable, as in the case of data from the cultured calls and animals (means ± standard error).
Reviewer 2 Report
I wondered whether the binding of the polymer to CD44 induces a CD44e dependent reaction. And: CD44 is an almost ubiquitously expressed protein. How do the authors think to reach specificity and to avoid uptake of the drugs into many non-tumor cells. Have they considered to expand the nano-polymers to interact wiith other surface proteins.
Reviewer 3 Report
Dear editor,
The manuscript is very good and perfect for publication. I have only one minor comment
The author should examine the delivery system for each analyte alone and in combination to give a clear idea about the mechanism and the potency of combination and the importance of the delivery system
With my best regards:
